# Toxin/antitoxin systems induce persistence and work in concert with restriction/modification systems to inhibit phage

Laura Fernández-García,[1,2] Sooyeon Song,[1,3,4] Joy Kirigo,[1] Michael E. Battisti,[1] Maiken E. Petersen,[1,5] María Tomás,[2] Thomas K. Wood[1]

**ABSTRACT** Myriad bacterial anti-phage systems have been described and often the mechanism of programmed cell death is invoked for phage inhibition. However, there is little evidence of "suicide" under physiological conditions for these systems. Instead of death to stop phage propagation, we show here that persister cells, i.e., transiently-tolerant, dormant, antibiotic-insensitive cells, are formed and survive using the *Escherichia coli* C496_10 tripartite toxin/antitoxin system MqsR/MqsA/MqsC to inhibit T2 phage. Specifically, MqsR/MqsA/MqsC inhibited T2 phage by $10^5$-fold and reduced T2 titers by 3,000-fold. During T2 phage attack, in the presence of MqsR/MqsA/MqsC, evidence of persistence includes the single-cell physiological change of reduced metabolism (via flow cytometry), increased spherical morphology (via transmission electron microscopy), and heterogeneous resuscitation. Critically, we found restriction-modification systems (primarily EcoK McrBC) work in concert with the toxin/antitoxin system to inactivate phage, likely while the cells are in the persister state. Hence, a phage attack invokes a stress response similar to antibiotics, starvation, and oxidation, which leads to persistence, and this dormant state likely allows restriction/modification systems to clear phage DNA.

**IMPORTANCE** To date, there are no reports of phage infection-inducing persistence. Therefore, our results are important since we show for the first time that a phage-defense system, the MqsRAC toxin/antitoxin system, allows the host to survive infection by forming persister cells, rather than inducing cell suicide. Moreover, we demonstrate that the MqsRAC system works in concert with restriction/modification systems. These results imply that if phage therapy is to be successful, anti-persister compounds need to be administered along with phages.

**KEYWORDS** phage inhibition, persistence, toxin/antitoxin systems

P hage attack, resulting in up to 40% bacterial cell death in oceans (1), is the universal constraint on bacterial immortality; hence, bacteria have developed numerous phage-inhibition systems. For phage inhibition, there is little evidence that these systems cause cell death; instead, it is much more likely these systems invoke dormancy (2, 3). Bacterial cell dormancy has several potential benefits: (i) allowing time for other host-encoded, phage-inhibition systems to function, (ii) slowing viral replication, and (iii) accommodating spacer accumulation for CRISPR-Cas (2). In fact, the very existence of spacers in CRISPR-Cas type III and VI systems; i.e., those shown to induce dormancy by targeting host RNA, demonstrate cells must survive phage attack to obtain spacers. Also, growth-inhibited cells have more spacer acquisition (4). In contrast, programmed cell death during a phage attack is paradoxical from the perspective of individual cells and has yet to be proven beneficial for kin.

Given that cells enter a state of dormancy known as persistence to survive numerous stresses [e.g., antibiotics (5), starvation (6), oxidation (7)] through guanosine

Address correspondence to Thomas K. Wood, tuw14@psu.edu.

The authors declare no conflict of interest.

See the funding table on p. 12.

pentaphosphate and tetraphosphate (p)ppGpp signaling (8), which leads to dimerization of their ribosomes to cease translation (9–11), we reasoned that phage attack is another form of extreme stress, so it should lead to persistence, in some cases. Also, phage-inhibition systems include toxin proteins of toxin/antitoxin systems, which are known to induce persistence by up to 14,000-fold (7, 12, 13). For example, after transcription shutoff by T4 phage activates the Hok/Sok toxin/antitoxin system, Hok damages the cell membrane to cease cell metabolism to halt phage propagation, in the first link of a toxin/antitoxin system to phage inhibition (1996) (14). Similarly, phage attack activates various toxins to reduce cell metabolism including those of the CBASS [phospholipases, nucleases, and pore-forming proteins (15)], RADAR [RdrB converts ATP to ITP (16)], and Thoeris [ThsA degrades $NAD^+$ (17)] phage inhibition systems.

Therefore, based on these two insights (extreme stress produces persisters and phage-inhibition systems utilize toxins to reduce metabolism), we investigated whether persister cells are formed as a result of inducing the *bona fide Escherichia* spp. phage inhibition system, the tripartite toxin/antitoxin/chaperone system *Escherichia coli* C496_10 MqsR/MqsA/MqsC (18, 19), which utilizes RNase MqsR (20, 21) upon phage attack. We confirmed that the MqsR/MqsA/MqsC, produced from its natural promoter, via the same system used to identify it, dramatically inhibits T2 phage (19), and discovered this system converts *E. coli* into the persister state to inhibit phage. In addition, we discovered that restriction/modification systems assist the MqsR/MqsA/MqsC toxin/antitoxin system to inhibit T2 phage. To our knowledge, this is the first connection of phage inhibition with persistence, and the first non-CRISPR-Cas system to be linked to restriction/modification systems.

## MATERIALS AND METHODS

### Bacteria and growth conditions

Bacteria and plasmids are shown in Table S1, and cells were cultured at 37°C in LB supplemented with 30 µg/mL of chloramphenicol (to retain plasmids, LB Cm30). The complete MqsR/MqsA/MqsC [from *E. coli* C496_10 (19)] plasmid-based system (pCV1-*mqsRAC* and pCV1 empty plasmid control) was verified via Plasmidsaurus sequencing.

### Persister formation assays, mitomycin C treatment, and phage titers

Single colonies were cultured overnight in LB-chloramphenicol, then diluted 100× and grown to a turbidity of 0.5 at 600 nm. T2 phage was then added (multiplicity of infection ~0.1) for 1 h, then the cells were washed twice with phosphate-buffered saline (PBS, 8 g of NaCl, 0.2 g of KCl, 1.15 g of $Na_2HPO_4$, and 0.2 g of $KH_2PO_4$ in 1,000 mL of $dH_2O$) and cultured for 3 h with LB-ampicillin [100 µg/mL, 10 minimum inhibitory concentration (MIC)] or LB-ciprofloxacin (5 µg/mL, 10 MIC) to kill non-persister cells (11). To verify persister cells were formed during ampicillin and T2 treatments, kill curves were generated by treating late exponential cells (turbidity at 600 nm of ~0.8) with 100 µg/mL ampicillin (10 MIC) for 4 h with shaking at 250 rpm or by treating late exponential cells (turbidity at 600 nm of ~0.8) with T2 phage at 0.1 MOI. Samples were taken every 30 min, washed with PBS, and 100 µL was serially diluted with PBS to determine the number of viable cells. Mitomycin C was added in LB (10 µg/mL, 5 MIC) to kill persister cells (22). Cells were washed twice with PBS and enumerated on LB-Cm plates to measure persistence. To determine phage titers, after 30 min of phage contact, supernatants were diluted with phage buffer (0.1 M NaCl, 10 mM $MgSO_4$, and 20 mM Tris-HCl pH 7.5), and plaques were enumerated on top agar, TA, (1% tryptone, 0.5% NaCl, 1.5% agar lower layer and 0.4% agar top layer) double layer plates.

### Persister resuscitation assay

Single-cell microscopy of resuscitating persister cells was performed as described previously (23) using LB agarose gel pads that were observed up to 3 h via a light

microscope (Zeiss Axioscope.A1, bl_ph channel at 1,000 ms exposure). The persister cells were generated by the addition of T2 phage at 0.1 MOI for 1 h. The microscope was maintained at 37°C by placing it in a vinyl glove box (Coy Labs), which was warmed by an anaerobic chamber heater (Coy Labs, 8535-025).

## Metabolic activity via flow cytometry

To ascertain metabolic activity, *E. coli* MG1655/pCV1-*mqsRAC* was grown to a turbidity of 0.5 at 600 nm, and T2 phage was added (MOI 0.1) for 1 h. Cells were washed, resuspended in PBS, stained for metabolic activity (BacLight RedoxSensor Green Vitality Kit, Thermo Fisher Scientific Inc., Waltham, MA, USA), which measures the cellular redox state. The fluorescence signal of 100,000 cells was analyzed by flow cytometry (Beckman Coulter LSRFortessa) using the FL1 (530 ± 30 nm).

## Transmission electron microscopy

To confirm persister cell formation, *E. coli* MG1655/pCV1-*mqsRAC* was grown to a turbidity of 0.5 at 600 nm, and T2 phage was added (MOI 0.1) for 1 h. Cells were centrifuged and fixed with 2% glutaraldehyde in 0.1 M sodium cacodylate buffer followed by secondary fixation with 1% osmium tetroxide. Samples were then stained with 2% uranyl acetate, dehydrated, and embedded in eponate. After curing, blocks were sliced with an ultramicrotome, and 70 nm thin slices were collected on grids and post-stained with uranyl acetate and lead citrate.

## Quantitative real-time reverse-transcription PCR (qRT-PCR)

To quantify transcription after T2 phage attack (15 min at MOI 0.1) in the presence of MqsR/MqsA/MqsC, RNA was isolated by cooling rapidly using ethanol/dry ice in the presence of RNA Later (Sigma) using the High Pure RNA Isolation Kit (Roche). The following qRT-PCR thermocycling protocol was used with the iTaq universal SYBR Green One-Step kit (Bio-Rad): 95°C for 5 min; 40 cycles of 95°C for 15 s, 60°C for 1 min for two replicate reactions for each sample/primer pair. The annealing temperature was 60°C for all primers. Primers are shown in Table S2.

## Efficiency of plating and efficiency of center of infection assays

The efficiency of plating (EOP) assay (24) was performed for *E. coli* by serially diluting phage T2 (dilutions from $10^{-1}$ to $10^{-7}$) onto double agar plates that were inoculated with 100 µL of overnight cultures and dried for a minimum 15 min. Diluted phage (5 µL) was added to soft agar plates, the plates were incubated overnight, and the number of plaque-forming units (PFU)/mL was determined. The efficiency of the center of infection (ECOI) assay (25) was used to determine the ability of T2 phage to infect different strains (i.e., a pre-adsorbed productive infection assay) and performed by diluting overnight bacterial cultures 1:100 into 15 mL of fresh buffered LB Cm30 and incubating to turbidity at 600 nm of ~0.5, then T2 phage (MOI ~0.1) was added to the cultures and incubated for 8 min (adsorption time). The cultures were washed twice with PBS (centrifuging at 5,000 rpm for 10 min) to remove free phages. Then the cultures were diluted in phage buffer (dilutions 1 to 5), and 5 µL drops were added to soft agar plates containing MG1655. Plates were placed at 37°C for overnight incubation and PFU/mL was determined.

## RESULTS

Given the extensive experience with *E. coli* persister cells (7, 9, 22, 26–31) and given our discovery of the MqsR/MqsA toxin/antitoxin system (20, 21), we chose a phage exclusion system of *E. coli* C496_10, the tripartite system MqsR (RNase toxin)/MqsA (antitoxin)/MqsC (SecB-type chaperone) (19), to investigate the hypothesis that persister cells are formed upon phage attack. We also used the same expression system (natural

promoter) that was used to demonstrate phage inhibition (via a plaquing assay). We focused on T2 phage infection since this phage was inhibited best by the MqsR/MqsA/MqsC system (19) and produced MqsR/MqsA/MqsC in a background that lacks this tripartite system (*E. coli* K-12).

## MqsR/MqsA/MqsC inhibits T2 phage

As published previously (19), T2 phage was efficiently inhibited by the MqsR/MqsA/MqsC TA system in our hands; i.e., MqsR/MqsA/MqsC increased cell viability by $10^5$-fold compared to an empty plasmid control (Fig. 1; Table S3). Corroborating the phage inhibition, the number of T2 phage decreased 500-fold in the presence of MqsR/MqsA/MqsC: $6 \pm 2 \times 10^5$ PFU/mL were obtained for MqsR/MqsA/MqsC whereas $2.6 \pm 0.5 \times 10^8$ PFU/mL were obtained for the empty plasmid when T2 was added initially at $7.14 \times$

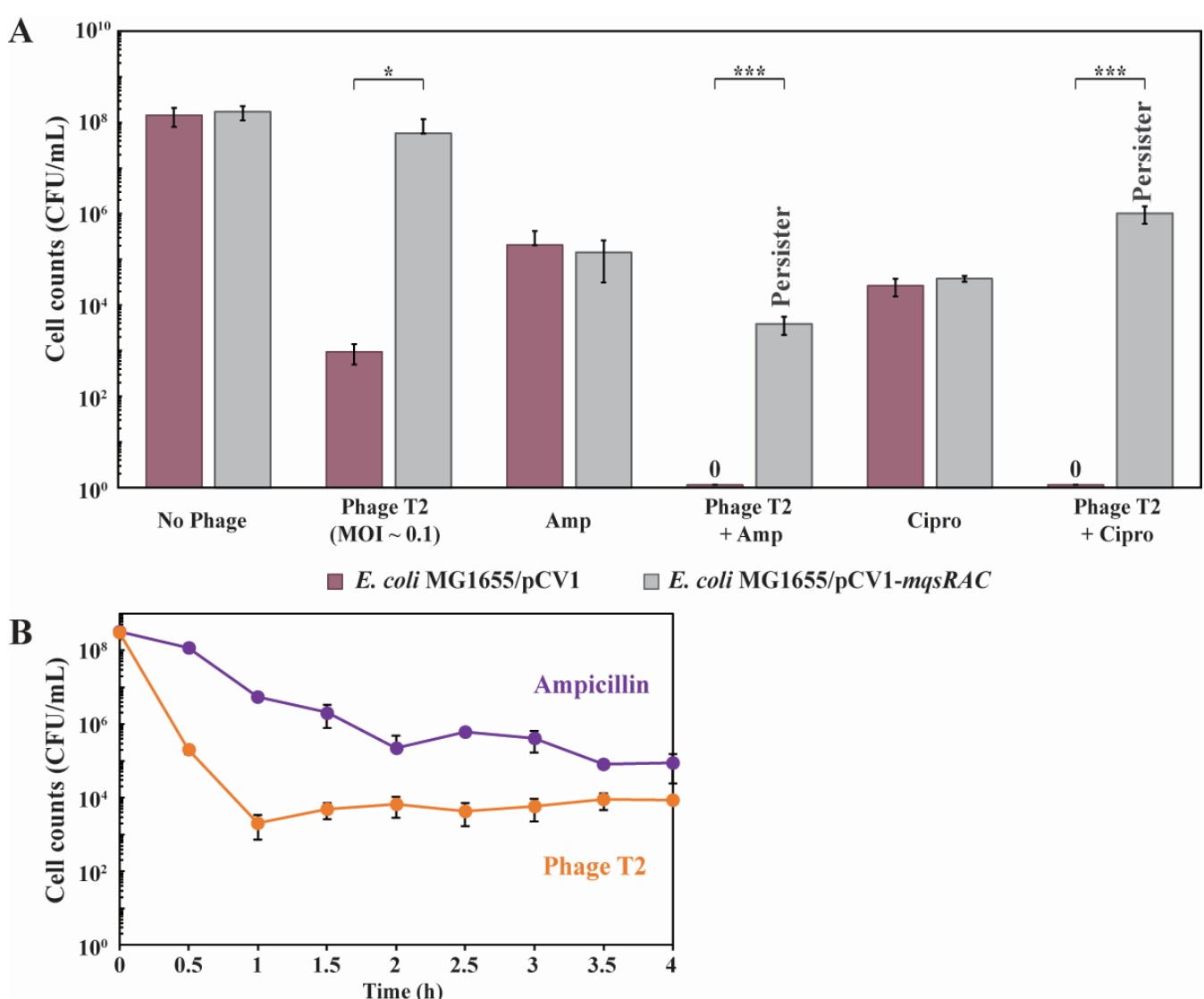

**FIG 1** MqsR/MqsA/MqsC inhibits T2 phage by forming persister cells. (A) Wildtype *E. coli* MG1655 cells producing MqsR/MqsA/MqsC from its native promoter (and the empty plasmid negative control) were contacted first with T2 phage at 0.1 MOI for 1 h followed by treatment with ampicillin ("Amp", 100 µg/mL, 10 MIC) or ciprofloxacin ("Cipro", 5 µg/mL, 10 MIC) for 3 h. Results from four independent cultures each. *** $P$-value < 0.005, **$P$-value < 0.01 (statistics via GradPath). One average deviation is shown, and raw data are shown in Table S3. (B) Kill curves with 10× MIC ampicillin (100 µg/mL) and T2 phage (0.1 MOI). Late exponentially-growing MG1655/pCV1 cells (turbidity 0.8) were treated at time 0 h.

$10^6$ PFU (MOI of 0.1). Therefore, MqsR/MqsA/MqsC actively inhibits T2 phage under these conditions.

To provide additional evidence of phage defense by the MqsR/MqsA/MqsC TA system, we performed both an EOP assay [to determine the number of phages present (24)] and an ECOI assay [to determine the number of pre-adsorbed phages (25)]. We found there was a 2,100 to a 5,000-fold reduction in the EOP with phage T2 when the cells produced MqsR/MqsA/MqsC (Fig. 2A; Table S4A). Consistently, we found the MqsR/MqsA/MqsC TA system reduces the ECOI by 5- to 14-fold (Fig. 2B; Table S4B). Hence, the MqsR/MqsA/MqsC TA system is a *bona fide* phage defense system.

## Cells surviving T2 phage attack with MqsR/MqsA/MqsC are persisters

Critically, treatment with T2 phage (0.1 MOI) for 1 h followed by treatment with 10× MIC of ampicillin for 3 h induced persistence in 0.1% of the cells producing MqsR/MqsA/MqsC (Fig. 1; Table S3); i.e., these cells survived treatment of lethal ampicillin concentrations. In contrast, cells bearing the empty plasmid were completely eradicated by ampicillin after exposure to T2 phage indicating no cells entered the persister state without MqsR/MqsA/MqsC (Fig. 1A; Table S3). Similar results were seen with a different class of antibiotic, ciprofloxacin (10× MIC), in that 2% cells with the MqsR/MqsA/MqsC phage inhibition system were persisters after T2 attack (Table S3), whereas no cells survived T2 and ciprofloxacin in the absence of MqsR/MqsA/MqsC (Table S3). Moreover, the increase in

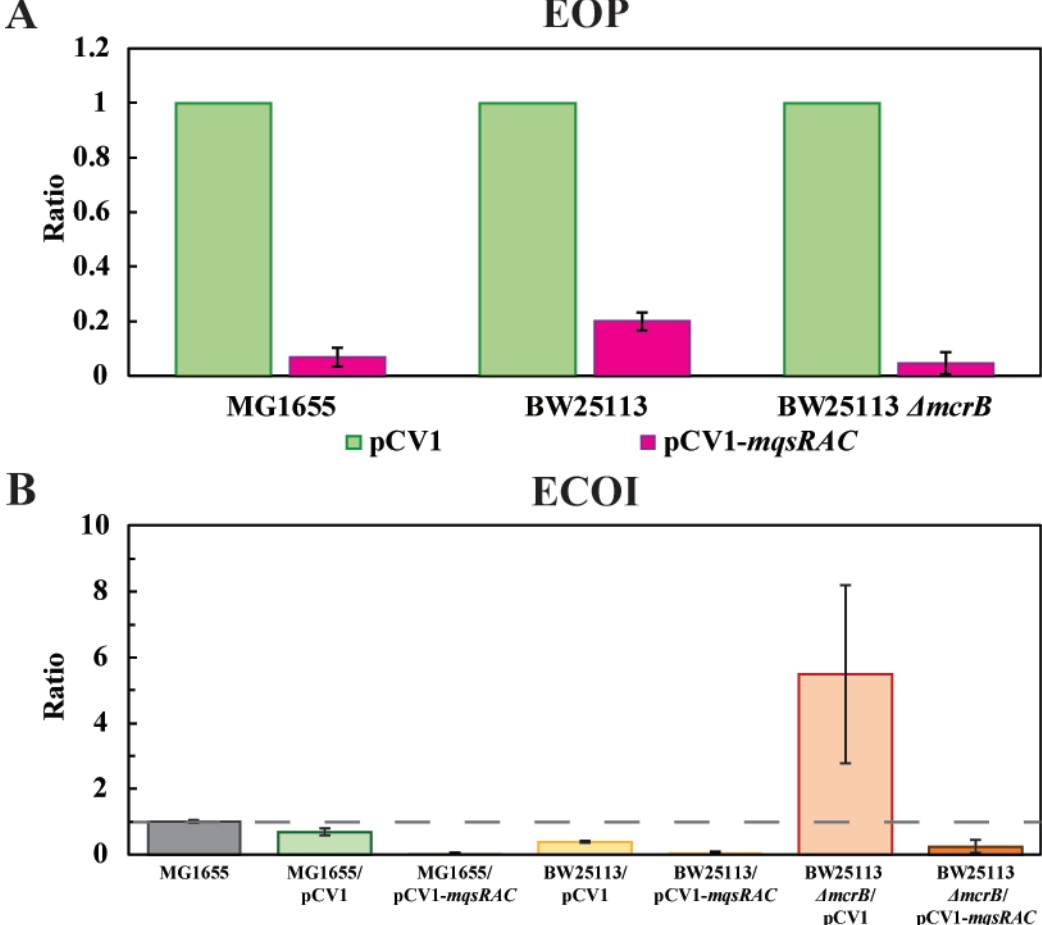

**FIG 2** EOP and ECOI assay. (A) EOP assay with light green columns showing results with empty plasmid pCV1 (no MqsRAC), and dark green columns showing results with MqsR/MqsA/MqsC (MqsRAC). (B) ECOI assay with the strains used in this manuscript. The efficiencies are relative to the strain MG1655. One average deviation is shown, and raw data are shown in Table S4.

persistence by MqsR/MqsA/MqsC was not due to a difference in growth since $mqsRAC^+$ had virtually no impact on the specific growth rate in the absence of phage (Fig. S1, μ = 1.7 ± 0.1/h, μ = 1.62 ± 0.06/h, and μ = 1.74 ± 0.09/h for MG1655, MG1655/pCV1, and MG1655/pCV1-*mqsRAC*, respectively).

To provide additional evidence of persistence with T2 phage, we tested MG1655/pCV1 cell viability with 100 µg/mL ampicillin (10 MIC) to generate a kill curve, and 2.5 h was found to be sufficient to form persister cells, which are indicated by the plateau in killing (Fig. 1B). Moreover, T2 phage infection also produced a plateau in killing, representative of the persister cells formed with ampicillin (Fig. 1B).

Since persister cells are metabolically inactive (5, 29, 32), we reasoned that if they are formed during a T2 phage attack, they must be resuscitated once the phage is removed and nutrients are supplied (23). Moreover, heterogeneity in persister cell resuscitation is a consistent, distinguishing feature of these cells (23, 33–36). Therefore, we investigated the resuscitation of single cells on LB gel pads after 1 h of T2 attack (0.1 MOI) and found significant heterogeneity in waking with four phenotypes including 41% with immediate waking (less than 30 min), 33% waking with a lag (30–180 min), 11% elongating, and 15% waking then dying (data from the representative sample) (Fig. 3; Table S5). In comparison, 97% ± 3% of the sample of exponentially-growing cells divide uniformly without elongation, lags, or death, forming microcolonies by 3 h (Fig. S2). Hence, during T2 attack, cells become persistent based on their antibiotic tolerance to two different classes of antibiotics, based on the formation of a plateau in killing, and based on their heterogeneity in resuscitation.

Further evidence of persistence after the T2 phage attack includes metabolic staining with flow cytometry, which revealed that after 1 h of T2 attack, *E. coli* cells with MqsR/MqsA/MqsC had significantly reduced metabolism compared to exponentially-growing cells (Fig. S3). This population with reduced metabolism has a broad range of metabolic activity since ~30% of the cells remain viable after T2 attack for MG1655/pCV1-*mqsRAC*) but only ~0.1% are dormant persisters (Table S3). In addition, transmission electron micrographs after the T2 phage attack show the same features of persister cells seen previously after lethal antibiotic treatment and after starvation: dense cytosol, intact membranes, and spheroid shape (6) (Fig. S4).

## Persistence cell formation is rapid upon phage attack

Bacteriophage T2 takes 40 min to complete infection, with a latent period of around 20 min and a rise period of 10 min (37). To investigate how rapidly persister cells form during T2 phage attack, we investigated the induction of the three loci required to inactivate ribosomes: *raiA, rmf,* and *hpf* (9) after 15 min of phage addition. Using qRT-PCR, we found no induction of these loci for the cells producing the MqsR/MqsA/MqsC phage inhibition systems (Fig. 4; Table S6), which indicates the cell probably uses existing proteins to become dormant rapidly, rather than relying on expression of ribosome inactivation genes once phage attacks. However, for cells that lack this phage inhibition system, *raiA* was induced fourfold at 15 min after phage attack, which indicates these cells are stressed as they unsuccessfully try to thwart T2 phage.

## Restriction/modification assists MqsR/MqsA/MqsC during persistence

Since cells with MqsR/MqsA/MqsC weather T2 attack by rapidly becoming persistent and resuscitating, we hypothesized that another system works while the cells are dormant to clear T2 phage. Therefore, we investigated whether cells that lack the restriction systems/DNA processing enzymes encoded by *mrr, mcrB,* and *recD* are susceptible to T2 attack and found that inactivation of the McrBC system was most important for defending against T2 since its inactivation reduces cell viability by 10,000-fold (Table 1). Critically, we found that the MqsR/MqsA/MqsC works with the McrBC restriction/modification system to defend against T2 phage since cells deficient in McrBC had 38-fold less viability, 33% less persister cells formed in the presence of MqsR/MqsA/MqsC (Fig.

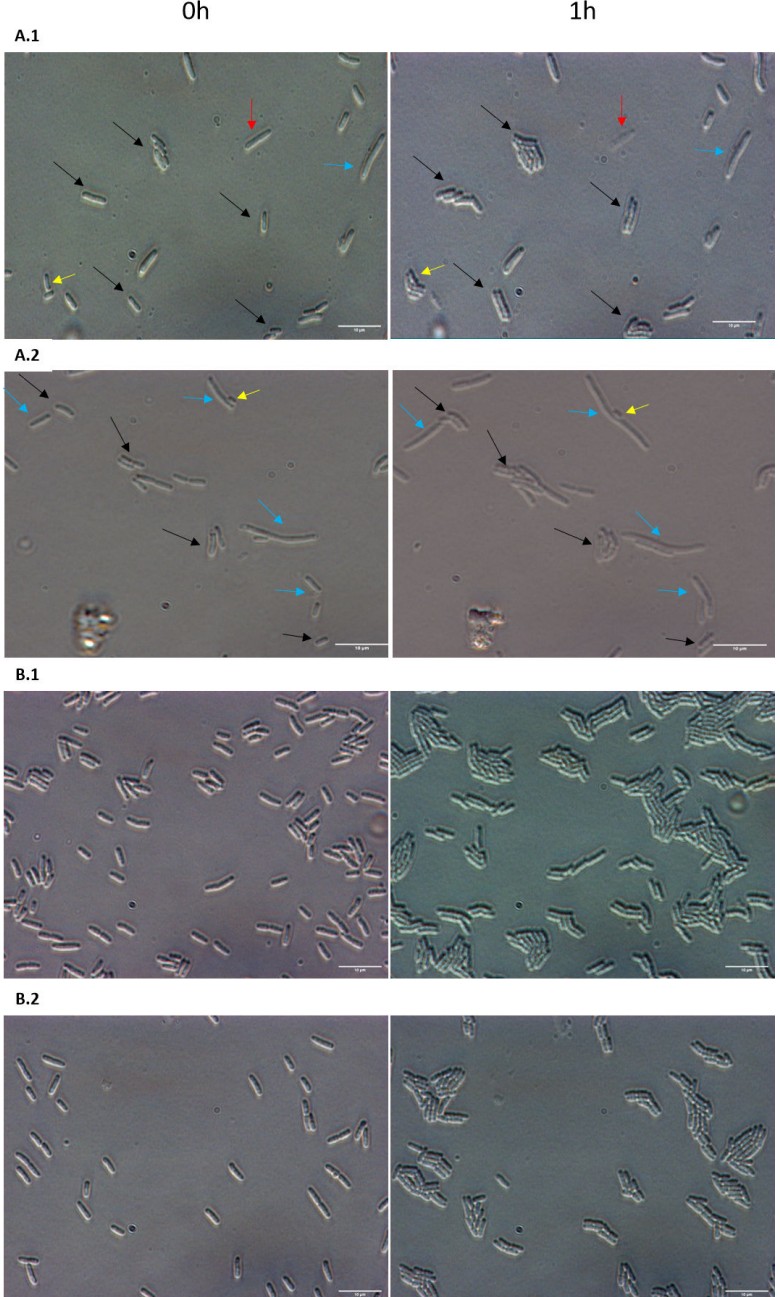

**FIG 3** Heterogenous single-cell resuscitation after phage attack. Representative images (from five independent cultures) of the resuscitation of *E. coli* persister cells with MqsR/MqsA/MqsC (A) and division of exponential cells (B) after 1 h as determined with light microscopy (Zeiss Axio Scope.A1) using LB agarose gel pads (representative images from 0 to 3 h are shown in Fig. S2). The persister cells were generated by the addition of T2 phage at 0.1 MOI for 1 h. Cells with the empty plasmid (i.e., no MqsR/MqsA/MqsC) are not shown due to the cellular debris that stems from complete eradication by T2 phage. Black arrows indicate cells with immediate waking (within 30 min), yellow arrows indicate cells with delayed waking (waking between 30 and 180 min), red arrows indicate cells that wake then die (lyse within 3 h), and blue arrows indicate cells that elongate initially. Black arrows were not added to exponential cultures pictures because all of them divide. Data for percentages are shown in Table S5.

5; Table 2), and a fourfold increase in PFU (Table 3). Moreover, in the absence of MqsR/MqsA/MqsC, there was a twofold reduction cell viability after the T2 phage attack when

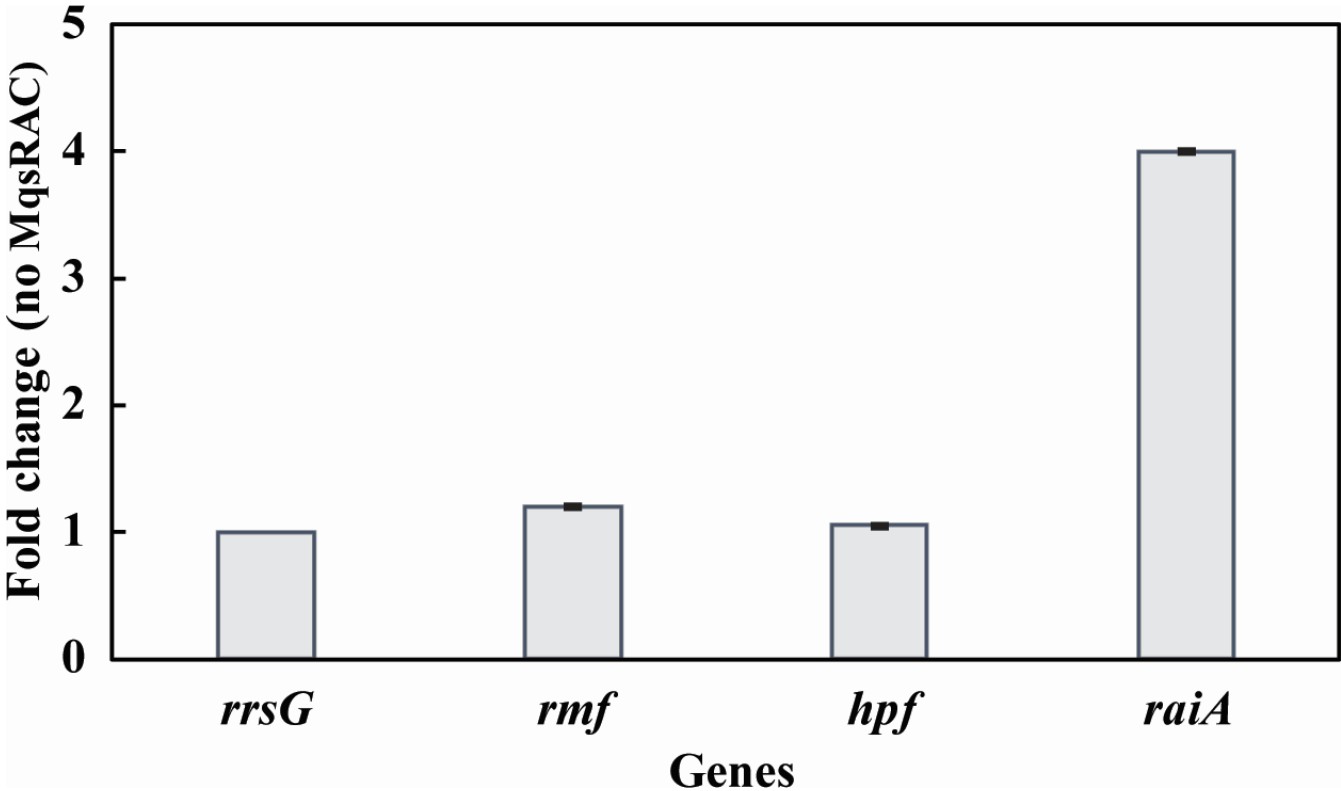

**FIG 4** Phage attack does not induce ribosome inactivation proteins. Fold changes for wild-type *E. coli* with the empty plasmid pCV1 (no MqsRAC) relative to the wild-type producing MqsR/MqsA/MqsC from pCV1-*mqsRAC*) after 15 min of T2 attack as determined by qRT-PCR. One average deviation is shown.

the McrBC restriction/modification systems were inactivated (Fig. 5; Table 2). Furthermore, in the presence of MqsR/MqsA/MqsC but in the absence of antibiotics, inactivating McrBC increases the production of T2 phage (PFU) by 84-fold as shown by the EOP assay and increases the ECOI by threefold (Table S4). Therefore, MqsR/MqsA/MqsC works in concert with restriction/modification systems to inhibit T2 phage.

## Mitomycin C kills *E. coli* phage-induced persister cells

Since phages are considered important for augmenting antibiotics to control infections (38), and since *E. coli* persister cells are formed rapidly during T2 phage attack, we investigated whether compounds that kill persister cells while they are dormant may be used to kill the remaining persisters. Approved by the FDA for cancer treatment, mitomycin C kills bacterial persister cells by passively diffusing into dormant cells and cross-linking their DNA (22) and has been shown to eradicate the pathogens *E. coli* O157:H7, *Staphylococcus aureus*, *Pseudomonas aeruginosa*, and *Acinetobacter baumannii* (22, 39). We found that adding mitomycin C (10 µg/mL, 5 MIC) to BW25113/pCV1-*mqsRAC* for 3 h after 1 h of T2 treatment (MOI 0.1) led to a 300,000-fold reduction in viable cells (Fig. 5; Table 2). Similar results were obtained when McrBC was inactivated

**TABLE 1** Survival after 30 min of T2 phage infection at 0.1 MOI[a]

| Strain | CFU/mL average |
|---|---|
| BW25113 | $1.4 \pm 0.3 \times 10^6$ |
| BW25113 Δ*mrr* | $8 \pm 2 \times 10^4$ |
| BW25113 Δ*recD* | $4.0 \pm 0.4 \times 10^6$ |
| BW25113 Δ*mcrB* | $2 \pm 1 \times 10^2$ |

[a]Includes results from two independent cultures.

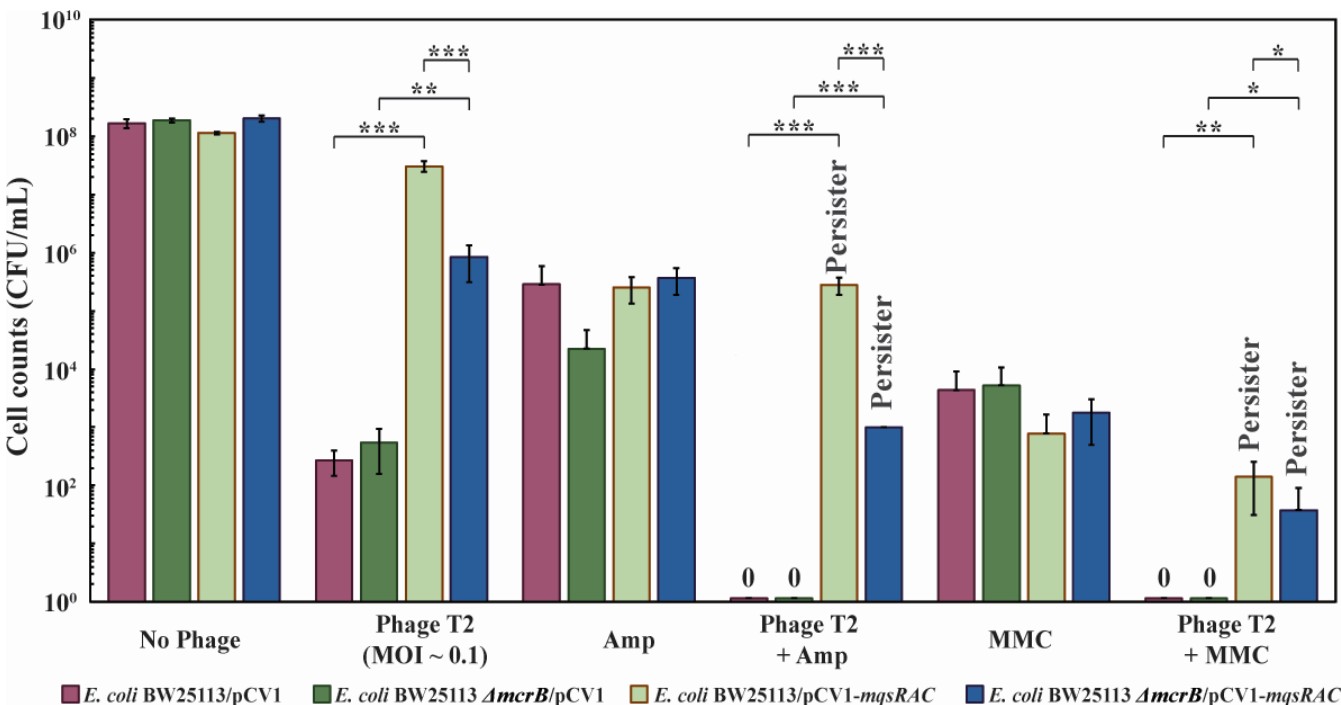

**FIG 5** MqsR/MqsA/MqsC inhibits T2 phage through cooperation with restriction/modification systems. Wildtype *E. coli* and Δ*mcrB* cells producing MqsR/MqsA/MqsC from its native promoter (and the empty plasmid negative control) were contacted first with T2 phage at 0.1 MOI for 1 h followed by treatment with ampicillin ("Amp", 100 µg/mL, 10 MIC) or mitomycin C ("MM C", 10 µg/mL, 5 MIC) for 3 h. Results from four independent cultures each. **** *P*-value < 0.0001, *** *P*-value < 0.005, ***P*-value < 0.01 (statistics via GradPath). One average deviation is shown.

(Fig. 5; Table 2). Hence, mitomycin C may be used to kill effectively the persister cells formed upon T2 phage attack.

## DISCUSSION

Our results demonstrate the tripartite toxin/antitoxin MqsR/MqsA/MqsC phage inhibition system forms persister cells upon T2 phage attack based on six lines of evidence: (i) multiple antibiotic tolerance (ciprofloxacin and ampicillin), (ii) heterogeneous resuscitation, (iii) metabolic inactivity via flow cytometry, (iv) morphology of persister cells via TEM, (v) plateau in killing, and (vi) MMC kills the MqsRAC-induced persister cells. Moreover, this persistence allows at least some of the cells to survive phage attack by providing time for phage DNA degradation by restriction/modification systems. Hence, our results suggest toxin/antitoxin systems act first to reduce metabolism, and then restriction/modification systems are utilized to undermine phage attacks. Similarly, CRISPR-Cas has been shown to act first, followed by restriction/modification systems in *Listeria* spp. (40). Our results showing synergy between toxin/antitoxin and restriction/modification systems are significant since both of these phage defense systems are found in nearly all bacteria [unlike many other phage inhibition systems such as CRISPR-Cas, CBASS, and retrons that are found in 39%, 13%, and 11% of procaryote genomes, respectively (41)], which suggests this interaction is widespread and may be dominant for phage inhibition. Our results also provide the first reproducible link between toxin/antitoxin systems and persister cells under near-physiological conditions [native promoter and copy number 5 vector used (19, 42)].

These results also have implications for other phage-inhibition systems in that, rather than causing cell death (43–46), these anti-phage systems probably induce dormancy; i.e., persistence, which allows the cell to thwart the phage in the same manner as we studied here. Along with the formation of persister cells, some cells indubitably succumb to phage attack; however, our results indicate the primary purpose of the anti-phage

TABLE 2 MqsR/MqsA/MqsC inhibits T2 phage through cooperation with restriction/modification systems[a]

| Strain | Condition | CFU/mL average | Survival % |
|---|---|---|---|
| BW25113/pCV1 | No Phage | $1.7 \pm 0.3 \times 10^8$ | 100% |
| | Phage T2 | $3 \pm 1 \times 10^2$ | $0.00017\% \pm 0.00007\%$ |
| | No phage + ampicillin | $3 \pm 3 \times 10^5$ | $0.2\% \pm 0.2\%$ |
| | No phage + mitomycin C | $4 \pm 5 \times 10^3$ | $0.003\% \pm 0.002\%$ |
| | Phage T2 + ampicillin | $0 \pm 0$ | $0\% \pm 0\%$ |
| | Phage T2 + mitomycin C | $0 \pm 0$ | $0\% \pm 0\%$ |
| BW25113 ΔmcrB/pCV1 | No Phage | $1.8 \pm 0.2 \times 10^8$ | 100% |
| | Phage T2 | $6 \pm 4 \times 10^2$ | $0.0003\% \pm 0.0002\%$ |
| | No phage + ampicillin | $2 \pm 2 \times 10^4$ | $0.01\% \pm 0.01\%$ |
| | No phage + mitomycin C | $5 \pm 5 \times 10^3$ | $0.002\% \pm 0.002\%$ |
| | Phage T2 + ampicillin | $0 \pm 0$ | $0\% \pm 0\%$ |
| | Phage T2 + mitomycin C | $0 \pm 0$ | $0\% \pm 0\%$ |
| BW25113/pCV1-mqsRAC | No Phage | $1.14 \pm 0.05 \times 10^8$ | 100% |
| | Phage T2 | $3.1 \pm 0.6 \times 10^7$ | $27\% \pm 6\%$ |
| | No phage + ampicillin | $3 \pm 1 \times 10^5$ | $0.2\% \pm 0.1\%$ |
| | No phage + mitomycin C | $8 \pm 9 \times 10^2$ | $0.0004\% \pm 0.0005\%$ |
| | Phage T2 + ampicillin | $2.8 \pm 0.9 \times 10^5$ | $0.9\% \pm 0.2\%$ |
| | Phage T2 + mitomycin C | $1 \pm 1 \times 10^2$ | $0.0004\% \pm 0.0005\%$ |
| BW25113 ΔmcrB/pCV1-mqsRAC | No Phage | $2.0 \pm 0.2 \times 10^8$ | 100% |
| | Phage T2 | $8 \pm 5 \times 10^5$ | $0.4\% \pm 0.2\%$ |
| | No phage + ampicillin | $4 \pm 2 \times 10^5$ | $0.2\% \pm 0.1\%$ |
| | No phage + mitomycin C | $2 \pm 1 \times 10^3$ | $0.0008\% \pm 0.0005\%$ |
| | Phage T2 + ampicillin | $1.0 \pm 0 \times 10^3$ | $0.6\% \pm 0.6\%$ |
| | Phage T2 + mitomycin C | $4 \pm 5 \times 10^1$ | $0.005\% \pm 0.007\%$ |

[a]BW25113 and BW25113 mcrB cells producing MqsR/MqsA/MqsC from its native promoter (and the empty plasmid negative control) were contacted first with T2 phage at 0.1 MOI for 1 h followed by treatment with ampicillin (10 MIC, 100 μg/mL) or mitomycin C (5 MIC, 10 μg/mL) for 3 h. Survival percentages are normalized based on no phage addition. Includes results from four independent cultures.

systems is not what has been termed "abortive infection" in which the phage inhibition system kills the cells (19, 43), but instead, these systems protect the cells and allow them to survive phage attack. Similar results showing dormancy (but not persistence) have been seen with type VI CRISPR-Cas and restriction/modification systems (40).

Corroborating our findings, many phage-inhibition systems reduce energy metabolites like ATP [RADAR (16) and Detocs (47)] and NAD$^+$ [Thoeris (17)], and the reduction of energy metabolites like ATP consistently induces persistence (11, 48). Furthermore, reduction of ATP by toxin/antitoxin systems leads to persistence (49).

Central to persister cell formation is an accumulation of (p)ppGpp (9, 10, 50, 51); hence, it is expected that if cells prevent phage propagation by becoming persisters, they will do so via an increase in (p)ppGpp. Therefore, to counter this bacterial defense, if our hypothesis/data are correct in this report, phages should try to reduce (p)ppGpp levels. Critically, this has been recently shown with *Pectobacterium* spp. phage PcCB7V, which encodes the nucleotide pyrophosphatase MazG that is enriched against TIR-STING anti-phage systems and reduces (p)ppGpp (52). In addition, these authors demonstrated that (p)ppGpp is increased during a phage attack (52). Additional evidence of the role of the inhibitory role (p)ppGpp during phage infection includes that lambda phage is not able to replicate with high (p)ppGpp concentrations due to (p)ppGpp-mediated

TABLE 3 Phage titers after 30 min of infection at 0.1 MOI[a]

| Strain | PFU/mL average | CFU/mL average |
|---|---|---|
| BW25113/pCV1-mqsRAC | $7 \pm 2 \times 10^5$ | $1.8 \pm 0.7 \times 10^8$ |
| BW25113 ΔmcrB/pCV1-mqsRAC | $2.9 \pm 0.9 \times 10^6$ | $3 \pm 1 \times 10^7$ |

[a]One average deviation shown. Includes results from four independent cultures.

inhibition of transcriptional activation of the lambda origin of replication (53) and that (p)ppGpp inhibits lambda $P_R$ by inhibiting formation of the first phosphodiester bond (54). Other (p)ppGpp-relevant phage results include that ppGpp stimulates lambda *paQ* promoter due to an increased rate of productive open complex formation (55) and that (p)ppGpp controls the lambda lysis-versus-lysogenization decision based on its differential influence of the activities of the lambda *pL, pR, pE, pI,* and *paQ* promoters (56).

Our model (Fig. 6) is that by invoking persistence (i.e., dormancy) through phage attack, backup phage inhibition systems like restriction modification systems can degrade the phage genome since restriction enzymes do not require any resources from the cell for activity; i.e., Gibbs free energy is less than zero for phage DNA degradation due to the increase in entropy, so degradation of the phage genome can occur while the cells sleep. This general concept of allowing backup DNA-targeting enzymes to function while the cell is dormant has been proposed for the type III/VI CRISPR-Cas systems (2, 57) and demonstrated while this work was being prepared for type VI CRISPR-Cas in *Listeria* spp. (40); however, the benefit of dormancy has not been suggested for the plethora of recently discovered, non-CRSPR-Cas phage inhibition systems. Here, we demonstrate that (i) dormancy in the form of persistence is a key aspect of phage inhibition of non-CRISPR-Cas systems and that (ii) restriction/modification is utilized during the dormant state.

Since phages rarely eradicate populations, a logical extension of our results is that for Bacteria and Archaea, phage attack should lead to a small population of surviving cells in the persister state, since only a small population of cells survive other stresses like starvation and antibiotics (58, 59). Hence, for medical applications, a combination

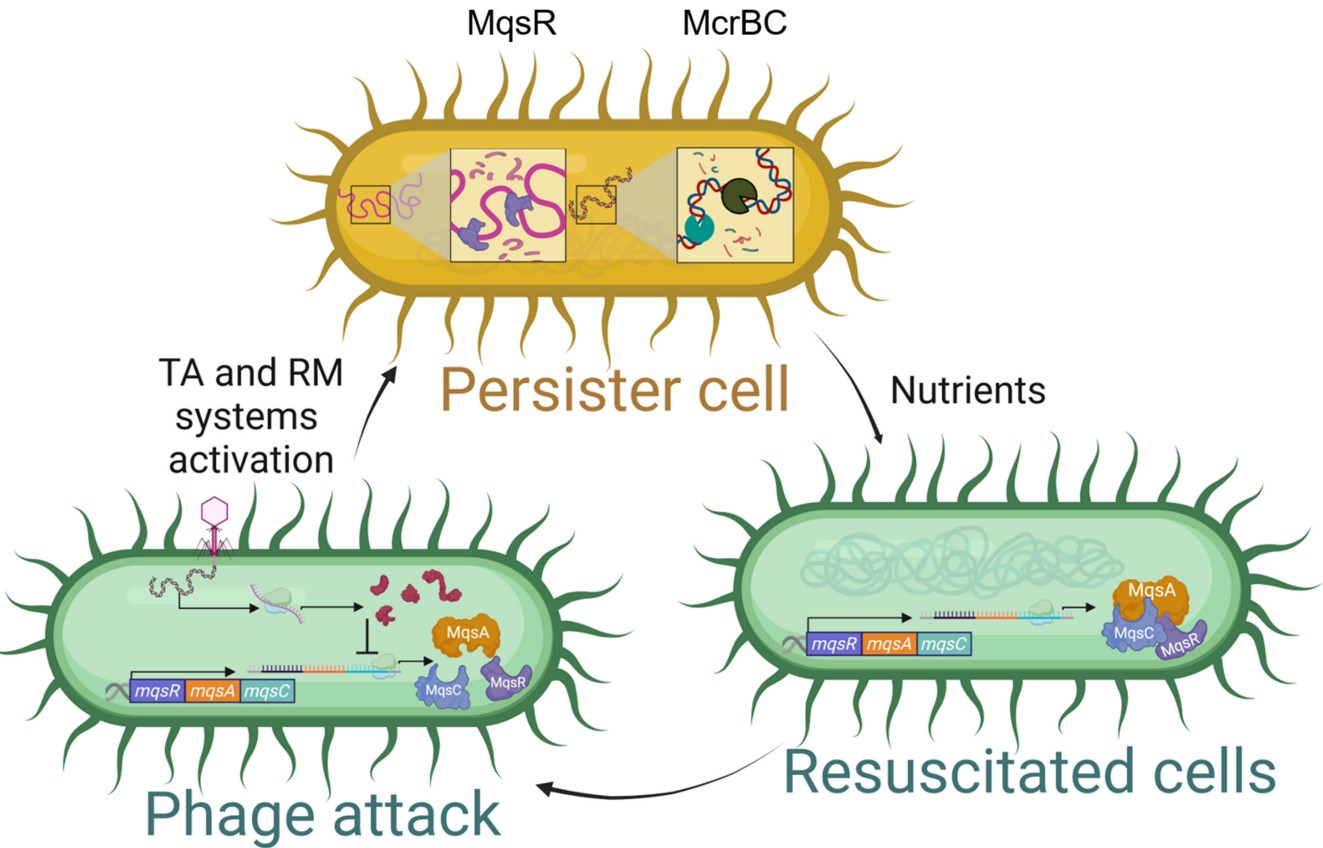

**FIG 6** Schematic illustrating phage inhibition systems invoke dormancy. Our results indicate that after phage attack, bacteria become persister cells (i.e., dormant), through RNase decay via toxin MqsR of the MqsR/MqsA/MqsC tripartite toxin/antitoxin system. During this period of inactivity, restriction/modification systems, which do not require cell resources, degrade phage DNA. Upon favorable conditions (nutrients provided), the persister cells resuscitate, having survived the phage attack.

treatment will likely be required that recognizes that persister cells will develop during phage therapy. Therefore, compounds such as mitomycin C, which is capable of killing a wide range of cells in the persister state (22), will likely be required, as we demonstrate (Fig. 5; Table 2).

Since the cells are becoming dormant (i.e., persisters) rather than dying, our results also suggest that after a phage attack, the cells that survive phage, resuscitate upon detecting nutrients, as we have shown using single-cell microscopy for persister cells (9, 10, 28), by using chemotaxis proteins and sugar transport membrane proteins, by reducing cAMP and ppGpp internal signals, and by increasing HflX activity to convert dimerized ribosomes (100S ribosomes) into active 70S ribosomes.

Finally, our results show that, like myriad other stresses such as starvation (6), oxidation (7), and antibiotic challenge (11), bacteria form persister cells to survive the extreme stress of phage attack. Hence, the formation of persister cells is a consistent and conserved means to weather stress.

## ACKNOWLEDGMENTS

We are grateful for the MqsR/MqsA/MqsC plasmids provided by Prof. Michael Laub and for assistance from Missy Hazen of the Microscopy and Cytometry team from The Huck Institutes of the Life Sciences at PSU.

This work was supported by both a Fulbright Scholar Fellowship and a Xunta de Galicia Postdoctoral Grant for L.F.G. and a National Research Foundation of Korea (NRF) grant from the Korean Government (NRF-2020R1F1A1072397) for S.Y.S. This study was also funded by grants PI22/00323 and PMP/00092 awarded to M.T. within the State Plan for R+D+I 2013–2016 (National Plan for Scientific Research, Technological Development and Innovation 2008–2011) and co-financed by the ISCIII-Deputy General Directorate for Evaluation and Promotion of Research—European Regional Development Fund "A way of Making Europe" and Instituto de Salud Carlos III FEDER.

## AUTHOR AFFILIATIONS

[1]Department of Chemical Engineering, Pennsylvania State University, University Park, Pennsylvania, USA

[2]Microbiology Department of Hospital A Coruña (CHUAC), Microbiology Translational and Multidisciplinary (MicroTM)-Research Institute Biomedical A Coruña (INIBIC) and University of A Coruña (UDC), A Coruña, Spain

[3]Department of Animal Science, Jeonbuk National University, Jeonju-Si, Jellabuk-Do, South Korea

[4]Agricultural Convergence Technology, Jeonbuk National University, Jeonju-Si, Jellabuk-Do, South Korea

[5]Interdisciplinary Nanoscience Center, Aarhus University, Aarhus, Denmark

## AUTHOR ORCIDs

Thomas K. Wood  http://orcid.org/0000-0002-6258-529X

## FUNDING

| Funder | Grant(s) | Author(s) |
| --- | --- | --- |
| Fulbright Scholar Fellowship | | Laura Fernández-García |
| Xunta de Galicia Postdoctoral Grant | | Laura Fernández-García |
| National Research Foundation of Korea (NRF) | NRF-2020R1F1A1072397 | Sooyeon Song |
| National Plan for Scientific Research | | María Tomás |

## AUTHOR CONTRIBUTIONS

Laura Fernández-García, Data curation, Formal analysis, Investigation, Methodology, Writing – review and editing | Sooyeon Song, Conceptualization, Data curation, Investigation | Joy Kirigo, Data curation, Investigation | Michael E. Battisti, Investigation | Maiken E. Petersen, Investigation | María Tomás, Supervision | Thomas K. Wood, Conceptualization, Formal analysis, Funding acquisition, Project administration, Resources, Supervision, Writing – original draft, Writing – review and editing

## DATA AVAILABILITY

The data underlying this article are available in the article and in online supplemental material.

## ADDITIONAL FILES

The following material is available online.

### Supplemental Material

**Supplemental material (Spectrum03388-23-s0001.docx).** Tables S1 to S6 and Fig. S1 to S4.

### Open Peer Review

**PEER REVIEW HISTORY (review-history.pdf).** An accounting of the reviewer comments and feedback.

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
