## [Reviewer comments · Microbiology Spectrum]

Microbiology Spectrum

Toxin/Antitoxin Systems Induce Persistence and Work in Concert with Restriction/Modification Systems to Inhibit Phage

Laura Fernández-García, Sooyeon Song, Joy Kirigo, Michael Battisti, Maiken Petersen, Maria Tomas, and Thomas Wood

Corresponding Author(s): Thomas Wood, The Pennsylvania State University

Review Timeline:

Submission Date:	September 18, 2023
Editorial Decision:	September 27, 2023
Revision Received:	October 5, 2023
Editorial Decision:	October 24, 2023
Revision Received:	November 3, 2023
Accepted:	November 6, 2023

Editor: M.-N. Frances Yap

Reviewer(s): The reviewers have opted to remain anonymous.

Transaction Report:

DOI: <https://doi.org/10.1128/spectrum.03388-23>

September 27, 2023

Prof. Thomas K. Wood
The Pennsylvania State University
Chemical Engineering
304 Chemical and Biochemical Engineering
UNIVERSITY PARK, Pennsylvania 16802

Re: Spectrum03388-23 (Toxin/Antitoxin Systems Induce Persistence and Work in Concert with Restriction/Modification Systems to Inhibit Phage)

Dear Dr. Wood:

Thank you for submitting your manuscript to Microbiology Spectrum. Two expert reviewers have provided opinions on your manuscript. The review comments are conflicting and also raise substantial gaps and shortcomings in data interpretation and experimental design. I will be happy to consider a revised manuscript provided that the concerns raised by the Reviewer #2 can be fully rebutted. Please refrain from submitting a revision and make a withdrawal request if the proposed experiments and logical inconsistencies can not be addressed within the next 60 days.

Link Not Available

Sincerely,

M.-N. Frances Yap

Journals Department
Reviewer comments:

Reviewer #1 (Comments for the Author):

The manuscript by Fernandez-Garcia provides evidence that toxin-antitoxin systems induce bacterial persistence and this persistence promotes cell survival after phage infection. The authors go on to show that in persister cells, restriction modification systems operate to protect 'sleeping' cells, presumably by destroying phage DNA. The authors frame their results in a way that contradicts the popular idea that phage defense systems protect cell populations from phage infection by killing the infected cell

before the phage can complete its life cycle, a mechanism analogous to apoptosis. This paper and others like it are important as the argument that bacteria persist and go dormant to survive phage infection rather than committing suicide is well reasoned and supported by data. I only have a few minor comments in no particular order.

Line 130: is MqsC not involved here?

Line 133: "...and produced MqsR/A/C..." This sentence is confusing. Do the authors mean T2 phage virions are produced?

Line 143: Describe briefly for your non-phage expert readers what the EOP and ECOI assays are.

Line 187: This sentence seems like it should be introduced at the beginning of the paragraph to help readers understand the timing of the phage lifecycle as it relates to the timing of persister formation.

Line 194: The inactivation of McrBC and its impact on phage replication is important and should be included as a main figure rather than a supplement.

Line 246: Inclusion of literature related to (p)ppGpp and lambda phage would be a welcome addition to the discussion:

Szalewska-Palasz A, Wegrzyn A, Herman A, Wegrzyn G. The mechanism of the stringent control of lambda plasmid DNA replication. *Embo J.* 1994;13(23):5779-85. Epub 1994/12/01. PubMed PMID: 7988574; PMCID: PMC395544.

Potrykus K, Wegrzyn G, Hernandez VJ. Multiple mechanisms of transcription inhibition by ppGpp at the lambda_{dap}(R) promoter. *J Biol Chem.* 2002;277(46):43785-91. Epub 20020910. doi: 10.1074/jbc.M208768200. PubMed PMID: 12226106.

Potrykus K, Wegrzyn G, Hernandez VJ. Direct stimulation of the lambda_{dapa}Q promoter by the transcription effector guanosine-3',5'-(bis)pyrophosphate in a defined in vitro system. *J Biol Chem.* 2004;279(19):19860-6. Epub 20040310. doi: 10.1074/jbc.M313378200. PubMed PMID: 15014078.

Slominska M, Neubauer P, Wegrzyn G. Regulation of bacteriophage lambda development by guanosine 5'-diphosphate-3'-diphosphate. *Virology.* 1999;262(2):431-41. doi: 10.1006/viro.1999.9907. PubMed PMID: 10502521.

Reviewer #2 (Comments for the Author):

This manuscript reports the results of experiments designed to shed light on the roles that bacteriophage T2 and the toxin antitoxin (TA) module MqsRAC play in tolerance to phage and antibiotic challenges. TA modules are ubiquitous in bacteria and evidence suggests that they play roles in stabilizing extra-genomic elements, inducing latency in response to stress and thwarting the development of phage. Some of the data supporting roles for TA systems remain controversial including the formation of so-called persisters. The presence of TA modules throughout the bacterial kingdom and their potential role in pathogenesis and tolerance to antibiotic treatment make their study highly significant. Here, the authors present confirmatory evidence that the MqsRAC module plays an important role in suppressing infection of *E. coli* by phage T2. The authors also explore the effects of the antibiotics ampicillin and ciprofloxacin on the antiphage activity of MqsRAC. The results are interesting, but the authors' conclusions are unclear and not supported by the data. The presentation is muddled as the authors claim that T2 causes the formation of persisters that protect against the drugs, or that the drugs cause persisters that are resistant to the phage. Moreover, the distinction between persister cells and those that are merely tolerant to stress is notoriously difficult to assess, and not done correctly in this case. The manuscript might be improved by attention to the clarity of the conclusions as well as the following issues.

1. Experiments are only done with T2, so the generality of the phenomena is unclear. The Laub lab showed that the Mqs system does not protect against T4, which is closely related to T2. If persistence is a general anti-phage protectant, it should work against another phage besides T2. This should be tested.
2. The authors state that they employed 10X the MIC of ampicillin to test for the ability of Mqs to induce tolerance to phage. But the results shown in Figure 1 and S3 reveal that treatment of MG1655 with 10 X MIC of Amp has no effect on cell growth, whether Mqs is expressed or not. The standard for identification of persisters is treatment that eliminates 99% of the population (Balaban, et al, 2019).
3. When challenged before Amp (or Cipro) treatment with T2, the Mqs- population is eradicated, suggesting a synergistic effect of the T2 and the drug (Fig. 1). This inhibition is significantly blocked by MqsRAC. The major question is why does T2 infection sensitize the cells to Amp (or Cipro) treatment? This is not consistent with the authors' conclusion that the formation of persisters by T2 and/or Mqs is the explanation for the findings.
4. The manuscript consistently contends that persisters are the result of T2 infection, but the effects measured require the presence of the Mqs plasmid. This is confusing.
5. Is it really a surprise that survival of cells to T2 is decreased by loss of one RM system? Same for MMC; of course, it will kill the cells whether or not they are non-growing.
6. The y-axis in Figure 4 has a misspelling (hcange).

Staff Comments:

Preparing Revision Guidelines

Please return the manuscript within 60 days; if you cannot complete the modification within this time period, please contact me. If you do not wish to modify the manuscript and prefer to submit it to another journal, please notify me of your decision immediately so that the manuscript may be formally withdrawn from consideration by Microbiology Spectrum.

Response for Spectrum 03388-23 “Toxin/Antitoxin Systems Induce Persistence and Work in Concert with Restriction/Modification Systems to Inhibit Phage”

We wish to thank the editor and two reviewers for their help with this manuscript and feel the manuscript is much-improved due to the careful review. We have addressed all the issues as indicated below (the Reviewers’ comments are underlined below and the changed text is highlighted yellow in the revised manuscript). Note the line numbers below refer to the *revised* text.

Specifically, we have conducted three sets of new experiments to better demonstrate that *E. coli* forms persister cells upon phage T2 attack: we show (i) ampicillin creates persister cells via a kill curve with MG1655/pCV1 (**Fig. 1B**), (ii) T2 phage makes persisters via a kill curve (6th line of evidence of persistence with phage) for MG1655/pCV1 + T2 (**Fig. 1B**), and (iii) confirmed amp has reduced killing based on a pCV1 plasmid feature (but kills the plasmid-free host as expected) (**Table R1** below). We also improved the figures (specifically **Fig. 1A**, **Fig. 2**, **Fig. 4**, **Fig. 5**, and **Fig. S1**).

Reviewer #1

The manuscript by Fernandez-Garcia provides evidence that toxin-antitoxin systems induce bacterial persistence and this persistence promotes cell survival after phage infection. The authors go on to show that in persister cells, restriction modification systems operate to protect 'sleeping' cells, presumably by destroying phage DNA. The authors frame their results in a way that contradicts the popular idea that phage defense systems protect cell populations from phage infection by killing the infected cell before the phage can complete its life cycle, a mechanism analogous to apoptosis. This paper and others like it are important as the argument that bacteria persist and go dormant to survive phage infection rather than committing suicide is well reasoned and supported by data. I only have a few minor comments in no particular order.

Thank you for your encouragement!

1. Line 130: is MqsC not involved here?

We agree and now list MqsC here (line 112) and apologize for the confusion. We discovered the binary (MqsR/MqsA) type II toxin/antitoxin system in *E. coli* K-12 in 2004 and characterized in subsequent years, but the current paper deals with a tripartite system (MqsR/MqsA/MqsA) from *E. coli* C496_10, identified by the Laub lab. To avoid confusion, we have added more detail to the Methods (line 57, where we now indicate the source), (line 112, where we indicate the function of each protein).

2. Line 133: "...and produced MqssR/A/C..." This sentence is confusing. Do the authors mean T2 phage virions are produced?

As suggested, we have made our points more clearly by updating this sentence in the first paragraph of the Results. First, we broke it into two sentences. Then, we added detail to explain what the ‘same expression system’ means (natural promoter, previously indicated on line 48 of the Intro) and explained better why we focused on T2 phage (since it was previously shown to be inhibited by MqsRAC) (line 115).

3. Line 143: Describe briefly for your non-phage expert readers what the EOP and ECOI assays are.

As suggested, we added text to explain the efficiency of plating assay (EOP) and the efficiency of center of infection (ECOI) (line 125):

“... we performed both an efficiency of plating (EOP) assay (to determine the number of phages present (1)) and an efficiency of the center of infection (ECOI) assay (to determine the number of pre-adsorbed phages (2)).”

4. Line 187: This sentence seems like it should be introduced at the beginning of the paragraph to help readers understand the timing of the phage lifecycle as it relates to the timing of persister formation.

As suggested, we moved this sentence (line 167). Thank you for this suggestion.

5. Line 194: The inactivation of McrBC and its impact on phage replication is important and should be included as a main figure rather than a supplement.

As suggested, we have moved **Table S7** to the main text (now **Table 1**).

6. Line 246: Inclusion of literature related to (p)ppGpp and lambda phage would be a welcome addition to the discussion:

Szalewska-Palasz A, Wegrzyn A, Herman A, Wegrzyn G. The mechanism of the stringent control of lambda plasmid DNA replication. *Embo J*. 1994;13(23):5779-85. Epub 1994/12/01. PubMed PMID: 7988574; PMCID: PMC395544.

Potrykus K, Wegrzyn G, Hernandez VJ. Multiple mechanisms of transcription inhibition by ppGpp at the lambda_{dap}(R) promoter. *J Biol Chem*. 2002;277(46):43785-91. Epub 20020910. doi: 10.1074/jbc.M208768200. PubMed PMID: 12226106.

Potrykus K, Wegrzyn G, Hernandez VJ. Direct stimulation of the lambda_{dapaQ} promoter by the transcription effector guanosine-3',5'-(bis)pyrophosphate in a defined in vitro system. *J Biol Chem*. 2004;279(19):19860-6. Epub 20040310. doi: 10.1074/jbc.M313378200. PubMed PMID: 15014078.

Slominska M, Neubauer P, Wegrzyn G. Regulation of bacteriophage lambda development by guanosine 5'-diphosphate-3'-diphosphate. *Virology*. 1999;262(2):431-41. doi: 10.1006/viro.1999.9907. PubMed PMID: 10502521.

Thank you very much for taking the time to point these out to us! We modified the Discussion to include these 4 refs as (text beginning line 235):

“Additional evidence of the role of the inhibitory role (p)ppGpp during phage infection includes that lambda phage is not able to replicate with high (p)ppGpp concentrations due to (p)ppGpp-mediated inhibition of transcriptional activation of the lambda origin of replication (3) and that (p)ppGpp inhibits lambda P_R by inhibiting formation of the first phosphodiester bond (4). Other (p)ppGpp-relevant phage results include that ppGpp stimulates lambda *paQ* promoter due to an increased rate of productive open complex formation (5) and that (p)ppGpp controls the lambda lysis-versus-lysogenization decision based on its differential influence of the activities of the lambda *pL*, *pR*, *pE*, *pI*, and *paQ* promoters (6).”

Reviewer #2

This manuscript reports the results of experiments designed to shed light on the roles that bacteriophage T2 and the toxin antitoxin (TA) module MqsRAC play in tolerance to phage and antibiotic challenges. TA modules are ubiquitous in bacteria and evidence suggests that they play roles in stabilizing extra-genomic elements, inducing latency in response to stress and thwarting the development of phage. Some of the data supporting roles for TA systems remain controversial including the formation of so-called persisters. The presence of TA modules throughout the bacterial kingdom and their potential role in pathogenesis and tolerance to antibiotic treatment make their study highly significant. Here, the authors present confirmatory evidence that the MqsRAC module plays an important role in suppressing infection of *E. coli* by phage T2. The authors also explore the effects of the antibiotics ampicillin and ciprofloxacin on the antiphage activity of MqsRAC. The results are interesting, but the authors' conclusions are unclear and not supported by the data. The presentation is muddled as the authors claim that T2 causes the formation of persisters that protect against the drugs, or that the drugs cause persisters that are resistant to the phage. Moreover, the distinction between persister cells and those that are merely tolerant to stress is notoriously difficult to assess, and not done correctly in this case. The manuscript might be improved by attention to the clarity of the conclusions as well as the following issues.

1. Experiments are only done with T2, so the generality of the phenomena is unclear. The Laub lab showed that the Mqs system does not protect against T4, which is closely related to T2. If persistence is a general anti-phage protectant, it should work against another phage besides T2. This should be tested.

The Laub group has already tested two phages (T2 and T4) and only found activity with T2, as indicated by the reviewer. However, we do not agree with the logic of the reviewer since if our hypothesis and data show persister cells are only made by MqsRAC, with no effect with the empty plasmid system, then we will only see effects with MqsRAC when it is actively inhibiting phage (and there is no dispute that MqsRAC inhibits T2 phage as the Laub group discovered this and we confirm it here). Hence, there will only be persisters formed when MqsRAC is active (i.e., RNase MqsR cleaves most mRNA and the cell becomes dormant), so there is no need to test other phages (e.g., T4) as MqsRAC is not active with T4 so no persisters should be formed and we will not glean much from this experiment (i.e., no different that the empty plasmid system). So, the requested experiment serves only as a negative control and we have already shown, with two antibiotics, that without MqsRAC but with T2 phage, there are no persisters formed (see T2 + Amp and T2 + cipro results in **Table S3**).

2. The authors state that they employed 10X the MIC of ampicillin to test for the ability of Mqs to induce tolerance to phage. But the results shown in Figure 1 and S3 reveal that treatment of MG1655 with 10 X MIC of Amp has no effect on cell growth, whether Mqs is expressed or not. The standard for identification of persisters is treatment that eliminates 99% of the population (Balaban, et al, 2019).

The reviewer makes an insightful observation, that 10X MIC kills less than expected for pCV1-bearing cells. First, 10X MIC kills ~90% of the pCV1-bearing cells, as shown in **Table 1/Table S3** in the text, rather than the expected survival of ~0.1% (so we disagree that ampicillin has ‘no effect’). This is one reason we confirmed our results with a second class of antibiotics, ciprofloxacin, which has expected killing with pCV1-based plasmids (0.2%, **Table S3**).

Second, as suggested, we re-examined the results of **Table S3** since the results with cells containing pCV1 have higher than expected ampicillin survival. As indicated previously in our results, we found that the presence of both plasmid pCV1 and pCV1-mqsRAC increase ampicillin resistance consistently by over 100X compared to the host MG1655 alone which has the expected 99.99% killing (100 µg/mL, 10 MIC ampicillin, for 3 h, OD = 0.5 so exponentially-growing cells, i.e., same conditions as in the paper):

Strain	Exponential (OD ₆₀₀ = 0.5)		
	% Survival	Std Dev	Fold Change
MG1655	0.01%	0.01%	1
MG1655/pCV1	4%	0.12%	385
MG1655/pCV1-mqsRAC	2%	0.04%	142

Table R1. pCV1 plasmid increases ampicillin resistance.

We added this information to the caption of **Table 1/Table S3** and clarified the text (line 136).

We had already sequenced both pCV1 and pCV1-mqsRAC (to know what we were using and ensure our results were dependent on MqsR/MqsA/MqsC) but unfortunately, we cannot determine why the plasmid increases Ap tolerance. There is nothing wrong with our approach as the host without the plasmid is killed in the expected manner with only 0.01% survival. To prevent confusion, we have explained this in the captions of **Tables S3/S7**. Note that pCV1 and pCV1-mqsRAC also consistently increased Amp tolerance in the stationary phase but stationary phase results with Amp are not as meaningful since Amp is less effective with slow-growing stationary-phase cells, of course. Furthermore, it is good we confirmed our results with ciprofloxacin to show phages induce persistence.

The plasmid map for our sequenced pCV1 is shown below (**Figure R1**). Therefore, our ampicillin has the expected killing for MG1655 (0.01% survival or 99.99% killing) and the only anomaly is that there is some pCV1-based factor, present for both pCV1 and pCV1-mqsRAC, that makes this system somewhat more tolerant to ampicillin. But again, we have confirmed all of our results with a separate antibiotic (cipro). And remember there is a 10⁸-fold phenotype in regard to survival with and without T2 phage with ampicillin treatment (**Fig. 1A**).

Figure R1. Plasmid map of pCV1.

Figure R2. Kill curves with 10X MIC ampicillin and T2 phage (MOI 0.1).

We also have performed an additional experiment with ampicillin, generating a kill curve, as indicated below.

3. When challenged before Amp (or Cipro) treatment with T2, the Mqs- population is eradicated, suggesting a synergistic effect of the T2 and the drug (Fig. 1). This inhibition is significantly blocked by MqsRAC. The major question is why does T2 infection sensitize the cells to Amp (or Cipro) treatment? This is not consistent with the authors' conclusion that the formation of persisters by T2 and/or Mqs is the explanation for the findings.

We disagree. The whole point is MqsRAC inhibits T2 phage by making persister cells. We previously provided not one but instead 5 lines of evidence showing persister cells are formed: (i) multiple antibiotic tolerance (ciprofloxacin and ampicillin), (ii) heterogeneous resuscitation, (iii) metabolic inactivity via flow cytometry, (iv) morphology of persister cells via TEM, and (v) MMC kills the MqsRAC-induced persister cells. Hence, our conclusion is sound that T2 fails to kill the MqsRAC-containing cells due to the formation of persister cells.

To provide a 6th line of evidence that persister cells are formed, we now show kill curves for MG1655/pCV1 + T2 (0.1 MOI) and for MG1655/pCV1 + 10 MIC Amp for more evidence (also addresses point #2 showing persisters form amp with pCV1) (Fig. R2 and in Fig. 1B in the text); hence, both T2 phage and ampicillin show the gold standard for persistence: a plateau in killing.

4. The manuscript consistently contends that persisters are the result of T2 infection, but the effects measured require the presence of the Mqs plasmid. This is confusing.

We apologize for the confusion. The main point of the manuscript is to demonstrate that a toxin/antitoxin system does not induce cell suicide (as claimed previously) but instead induces persistence. We have provided 6 lines of evidence to demonstrate our claim: (i) multiple antibiotic tolerance (ciprofloxacin and ampicillin), (ii) heterogeneous resuscitation, (iii) metabolic inactivity via flow cytometry, (iv) morphology of persister cells via TEM, (v) MMC kills the MqsRAC-induced persister cells, and (vi) kill curves show both ampicillin and T2 phage form persister cells. In addition, and perhaps the cause of the confusion, the MqsRAC system is not native to *E. coli* K-12 (the host we used for all the experiments) but instead is from *E. coli* C496_10; we now have tried to make this more clear (see lines 57 & 112 & 116).

5. Is it really a surprise that survival of cells to T2 is decreased by loss of one RM system?

Yes, especially since it is a completely novel result; i.e., it has never been shown before that TAs work with any other phage defense method. Moreover, of all phage defense systems, only CRISPR-Cas has been shown to work with restriction enzymes. So we find that not only does MqsRAC not work alone, we identified the other phage defense method with which it functions.

Same for MMC; of course, it will kill the cells whether or not they are non-growing.

We agree this result is straightforward, but its utility is that it provides additional proof for persistence as MMC is known to kill persister cells (7). Furthermore, we wish to demonstrate that phages alone will lead to persister cells and should be combined with anti-persister measures such as MMC (and we have shown here for the first time that the persister cells generated my TAs can be killed successfully with MMC). We note this is also novel and has not been shown previously.

6. The y-axis in Figure 4 has a misspelling (hchange).

Thank you very much for catching this! The y-axis has been corrected!

References in the Response

1. Kutter, E. (2009) In Martha R. J. Clokie, A. M. K. (ed.), *Bacteriophages: Methods and Protocols, Volume 1: Isolation, Characterization, and Interactions*. Humana Press, a part of Springer Science+Business Media, Vol. 501.
2. Sing, W.D. and Klaenhammer, T.R. (1990) Characteristics of phage abortion conferred in lactococci by the conjugal plasmid pTR2030. *Microbiology*, **136**, 1807-1815.
3. Szalewska-Pałasz, A., Węgrzyn, A., Herman, A. and Węgrzyn, G. (1994) The mechanism of the stringent control of lambda plasmid DNA replication. *The EMBO Journal*, **13**, 5779-5785.
4. Potrykus, K., Węgrzyn, G. and Hernandez, V.J. (2002) Multiple Mechanisms of Transcription Inhibition by ppGpp at the λ p R Promoter*. *Journal of Biological Chemistry*, **277**, 43785-43791.
5. Potrykus, K., Węgrzyn, G. and Hernandez, V.J. (2004) Direct Stimulation of the λ paQ Promoter by the Transcription Effector Guanosine-3',5'-(bis)pyrophosphate in a Defined in Vitro System*. *Journal of Biological Chemistry*, **279**, 19860-19866.
6. Słomińska, M., Neubauer, P. and Węgrzyn, G. (1999) Regulation of Bacteriophage λ Development by Guanosine 5'-Diphosphate-3'-diphosphate. *Virology*, **262**, 431-441.
7. Kwan, B.W., Chowdhury, N. and Wood, T.K. (2015) Combatting Bacterial Infections by Killing Persister Cells with Mitomycin C. *Environmental Microbiology*, **17** 4406–4414.

Re: Spectrum03388-23R1 (Toxin/Antitoxin Systems Induce Persistence and Work in Concert with Restriction/Modification Systems to Inhibit Phage)

Dear Dr. Wood,

While the reviews of the revised version are overall positive, a few important points by referee #3 are made that should be addressed prior to further consideration. Some of these can be addressed through text-only modifications, but others may require additional analysis and interpretation of the current data set. I will be more than happy to consider a revised manuscript that addresses these concerns.

Revision Guidelines

Sincerely,
M.-N. Frances Yap
Editor, Microbiology Spectrum

Reviewer #2 (Comments for the Author):

The authors have responded adequately to this reviewer's concerns.

Reviewer #3 (Comments for the Author):

In the recent years we saw a dramatic development in the field of antiphage immune systems. Many of these are operating at toxin-antitoxin (TA) modules and are commonly believed to act by promoting altruistic suicide of the infected cell, executing the so-called 'abortive infection'. However, the hard evidence for this mechanism is often not as hard as one would like it to be. The reason being that it is an experimentally challenging thing to directly observe. For one thing, when the phage infects the cell, it executes a hostile takeover program that disrupts normal cellular function, e.g. commonly the host chromosome is degraded (e.g. in the case of T2). Therefore, it is hard to say who killed the cell - was it the defence system? Or was the cell already killed by the phage and the defence system merely 'aborted the infection' in a sense that is commonly used by virologists studying the viruses of eukaryotes: aborted the efficient production of new viral particles, without implying the cell death.

Beta-lactam antibiotics such as ampicillin kill bacteria interfering with their cell wall synthesis. The efficiency of killing is proportional to the growth rate - the faster one grows, the more sensitive one is. However, bactericidal antibiotics (such as beta-lactams) do not sterilise bacterial cultures, some cells survive. When only a small fraction survives, these are dubbed as persisters. When the whole population is killed slowly, this is referred to as tolerance. The distinction between the two can be fuzzy. Expression of any toxic protein that slows down the growth leads to tolerance / persistence. TA effectors are toxic, and many slow down the growth engaged - which has led to active research on TAs role in persistence. It is a very controversial research topic muddled with botched up experimental systems, overinterpretation and hype.

In this study Fernández-García characterise a recently discovered antiphage toxin-antitoxin-chaperone (TAC) system MqsRAC that protects *E. coli* from T2. They characterise the interplay between MqsRAC and the classical antiphage systems: restriction systems/DNA processing enzymes and probe the connections between antibiotic persistence, MqsRAC and phage attack/defence.

As MqsRAC is triggered upon sensing the phage, it would compromise the bacterial growth, thus driving bacterial antibiotic tolerance. This is exactly what the authors see. I am less sure about the biological meaningfulness, really.

Note that for bacteria to 'persist' after the T2 attack, the chromosome needs to survive the T2-exercised chromosomal digestion. Does this mean MqsRAC TAC is triggered early during infection... I strongly recommend doing DNA staining.

Importantly the authors see that MqsRAC and restriction modification act together to defeat the phage attack. To my mind this is the most interesting result of the paper. However, these experiments (correct me if I am wrong) are only presented in the presence of ampicillin. I would really like to see these assays done in the absence of the antibiotic, with focus on PFUs, not CFUs. As it stands, the most biologically meaningful result is not fully exploited / documented.

Overall, while I think the data is interesting and thought-provoking, I am not sure it is a fully biologically watertight.

Response for Spectrum 03388-23R1 “Toxin/Antitoxin Systems Induce Persistence and Work in Concert with Restriction/Modification Systems to Inhibit Phage”

We wish to thank the editor and three reviewers for their help with this manuscript and feel the manuscript is improved due to the careful review. We have addressed all the issues as indicated below (our comments are underlined below and the changed text is highlighted yellow in the revised manuscript). Note the line numbers below refer to the *revised* text.

Specifically, we have conducted one additional experiment to determine if MqsRAC protects the host cDNA during T2 infection (but did not find evidence for this). We note that the PFU values requested by R3 were already reported in **Table S4** for the effect of phage in the absence of antibiotics as explained below, and we now make these results more obvious in the Results section (line 186).

Critically, we also re-conducted nearly all the antibiotic-related work of **Fig. 1** and **Fig. 5** and **Table 1B** and **Table S3** to confirm the antibiotic treatment results. From these new experiments, we found the expected, i.e., greater, killing of *both* the host and plasmid-bearing strains (99.99%) with a new batch of 10 MIC ampicillin. The main results/conclusions remain the same, and the artifact of 90% killing with ampicillin (earlier drafts) vs. 99.99% killing (current draft) has been removed from all the data now, so that we now show 0.2% persisters with T2 + amp treatment rather than 8%, which is line with our previous and others work (line 133). Hence, all the conclusions and main results have been confirmed and checked with at least 8 independent cultures. We apologize for the artifact of the ampicillin killing previously (the ampicillin killing on the host and two plasmid-bearing strains was checked by an undergraduate previously, which led us to conclude that the plasmids had some amp-resistance gene but this was not the case, hence, we repeated all the ampicillin experiments).

Reviewer #2

The authors have responded adequately to this reviewer's concerns.

Thank you for all of your help with this manuscript!

Reviewer #3

In the recent years we saw a dramatic development in the field of antiphage immune systems. Many of these are operating at toxin-antitoxin (TA) modules and are commonly believed to act by promoting altruistic suicide of the infected cell, executing the so-called 'abortive infection'. However, the hard evidence for this mechanism is often not as hard as one would like it to be. The reason being that it is an experimentally challenging thing to directly observe. For one thing, when the phage infects the cell, it executes a hostile takeover program that disrupts normal cellular function, e.g. commonly the host chromosome is degraded (e.g. in the case of T2). Therefore, it is hard to say who killed the cell - was it the defence system? Or was the cell already killed by the phage and the defence system merely 'aborted the infection' in a sense that is commonly used by virologists studying the viruses of eukaryotes: aborted the efficient production of new viral particles, without implying the cell death.

Beta-lactam antibiotics such as ampicillin kill bacteria interfering with their cell wall synthesis. The efficiency of killing is proportional to the growth rate - the faster one grows, the more sensitive one is. However, bactericidal antibiotics (such as beta-lactams) do not sterilise bacterial cultures, some cells survive. When only a small fraction survives, these are dubbed as persisters. When the whole population is killed slowly, this is referred to as tolerance. The distinction between the two can be fuzzy. Expression of any toxic protein that slows down the growth leads to tolerance / persistence. TA effectors are toxic, and many slow down the growth engaged - which has led to active research on TAs role in persistence. It is a very controversial research topic muddled with botched up experimental systems, overinterpretation and hype.

In this study Fernández-García characterise a recently discovered antiphage toxin-antitoxin-chaperone (TAC) system MqsRAC that protects *E. coli* from T2. They characterise the interplay between MqsRAC and the classical antiphage systems: restriction systems/DNA processing enzymes and probe the connections between antibiotic persistence, MqsRAC and phage attack/defence.

As MqsRAC is triggered upon sensing the phage, it would compromise the bacterial growth, thus driving bacterial antibiotic tolerance. This is exactly what the authors see. I am less sure about the biological meaningfulness, really.

Antibiotic tolerance (such as tolerance to beta-lactams due to slower growth, as indicated above) is distinct from persistence (as pointed out by the reviewer). Our paper shows for the first time that phage attack causes persistence; i.e., beyond what is expected for a slow-growth phenotype, we show antibiotic tolerance to two distinct antibiotics, including one that kills slow-growing bacteria but not persists (i.e., cipro) along with heterogeneous resuscitation, etc., so we demonstrate not just slow growth but persistence, which we feel is highly biologically-relevant.

Moreover, we agree both the TA field and the persistence field are botched up, primarily due to (i) poorly-designed experiments such as those confusing cells undergoing a stress response with those in the persister state and (ii) experiments with over-produced toxins, with authors claiming that this is physiologically-relevant. We note that 27 years ago, we discovered TAs inhibit phage (using Hok/Sok and several phage including T4), and got the mechanism right in the first paper relating TAs to phage (transcription shutoff), so along with DNA stabilization, we discovered, then, the primary physiologically-meaningful role of TAs that nearly everyone agrees about now, so we have some history of getting it right biologically (other examples include discovering the exporter of AI-2 and the role of indole as an interkingdom signal in the gut). We also have the only molecular model for persister formation and resuscitation, again, getting it biologically correct. Similarly, here, we show many are wrong in their assumption of cell suicide during phage defense. Therefore, these two discoveries in this one manuscript, (i) persistence due to phage attack and (ii) lack of suicide during phage attack, are certainly important biologically (along with (iii) the first report of a non-CRSPR anti-phage system using another anti-phage system; i.e., TAs + a restriction/modification system here).

1. Note that for bacteria to 'persist' after the T2 attack, the chromosome needs to survive the T2-exercised chromosomal digestion. Does this mean MqsRAC TAC is triggered early during infection... I strongly recommend doing DNA staining.

As suggested, we investigated whether MqsRAC is triggered early or late during T2 infection by measuring cDNA levels with a fluorescent dye (SYTO 9 green). Since T2 phage degrades 1/3 of host DNA in 20 min (1), we measured cDNA levels after T2 attack in the presence and absence of MqsRAC at 20 and 40 min (BW15113/pCV1-*mqsRAC* vs. BW15113/pCV1). Unfortunately, we found no difference in cDNA levels (please see **Table R1** below). For these experiments, we tried six independent cultures and utilized a Turner Designs fluorometer (TD-70) to measure fluorescence (indicating the amount of DNA) and a Fisher Scientific model 60 sonic dismembrator (1 min at power level 10 followed by centrifugation at 15,000 g for 5 min, and analysis of the supernatant).

2. Importantly the authors see that MqsRAC and restriction modification act together to defeat the phage attack. To my mind this is the most interesting result of the paper. However, these experiments (correct me if I am wrong) are only presented in the presence of ampicillin. I would really like to see these assays done in the absence of the antibiotic, with focus on PFUs, not CFUs. As it stands, the most biologically meaningful result is not fully exploited / documented.

We point out that these PFU results in the absence of ampicillin were previously presented in the EOP results (**Table S4A**) and ECOI results (**Table S4B**) and referred to on line 186 of the text so they have been provided twice previously. We found that the inactivation of McrB in the absence of antibiotic results in 84-fold more PFU as shown by the efficiency of plating assay (**Table S4A**, BW25113/pCV1-*mqsRAC* vs. BW25113 Δ *mcrB*/pCV1-*mqsRAC*) and 3-fold more phage as shown by the center of infection assay (**Table S4B**, BW25113/pCV1-*mqsRAC* vs. BW25113 Δ *mcrB*/pCV1-*mqsRAC*). So along with the 38-fold reduction in cell viability due to inactivation of McrB (in the absence of antibiotics) (**Table 1B**), we have shown clearly that McrB works with MqsRAC to thwart T2 infection.

To avoid confusion, we modified line 186 to make these results more clear and apologize for the confusion in the last draft.

Overall, while I think the data is interesting and thought-provoking, I am not sure it is a fully biologically watertight.

Table R1: Raw fluorescence values of cell lysates of *E. coli* BW25113/pCV1 and *E. coli* BW25113/pCV1-*mqsRAC* after T2 phage treatment for 20 and 40 min using SYTO™ 9 Green Fluorescent Nucleic Acid Stain.

Experiment 1: 1:3 dilution			
Strain	Time (min)	Fluorescence values	
		1	2
E. coli BW25113/pCV1	20	2315	2315
	40	2315	2315
E. coli BW25113/pCV1- mqsRAC	20	2318	2318
	40	2314	2314

Experiment 2: 1:100 dilution			
Strain	Time (min)	Fluorescence values	
		3	4
E. coli BW25113/pCV1	20	446	351
	40	416	440
E. coli BW25113/pCV1- mqsRAC	20	415	334
	40	420	415

Experiment 3: 1:100 dilution			
Strain	Time (min)	Fluorescence values	
		5	6
E. coli BW25113/pCV1	20	495	480
	40	487	491
E. coli BW25113/pCV1- mqsRAC	20	590	516
	40	589	520

REFERENCES

1. Hershey , A.D., Dixon , J. and Chase , M. (1953) Nucleic Acid Economy in Bacteria Infected with Bacteriophage T2 : I. Purine And Pyrimidine Composition. *Journal of General Physiology*, **36**, 777-789.

Re: Spectrum03388-23R2 (Toxin/Antitoxin Systems Induce Persistence and Work in Concert with Restriction/Modification Systems to Inhibit Phage)

Dear Dr. Wood:

Your manuscript has been accepted, and I am forwarding it to the ASM production staff for publication. Your paper will first be checked to make sure all elements meet the technical requirements. ASM staff will contact you if anything needs to be revised before copyediting and production can begin. Otherwise, you will be notified when your proofs are ready to be viewed.

Sincerely,
M.-N. Frances Yap
Editor
Microbiology Spectrum